

# Application of composite reference intervals in the diagnosis of subclinical hypothyroidism in the elderly: a retrospective study

Peijuan Li[1], Wenming Yang[1], Guohua Tang[2] and Zhipeng Li[1]

[1] General Practice Ward/International Medical Center Ward, General Practice Medical Center, West China Hospital, Sichuan University, Chengdu, China

[2] Health Management Center, General Practice Medical Center, West China Hospital, Sichuan University, Chengdu, China

## ABSTRACT

**Background**. Thyroid stimulating hormone releasing hormone (TSH) is a key indicator for diagnosing subclinical hypothyroidism (SCH). We evaluated factors affecting TSH levels in elderly SCH, establishing a composite reference interval, and comparing it with traditional one in diagnosis.

**Methods**. We collected data on patients aged ≥60 undergoing physical examinations in Chengdu, screening the influencing factors associated with TSH. Then, a two-dimensional composite reference interval was established for TSH and FT4, and the differences between the new and traditional diagnosing methods were compared.

**Results**. The incidence of subclinical thyroid dysfunction was about 14%, with SCH accounting for 97%. Regression analysis found that TSH levels increase as FT4 and uric acid levels decrease. Compared with the two-dimensional composite reference interval, the traditional one has a higher incidence rate of SCH.

**Conclusion**. Compared with the two-dimensional composite reference interval, the traditional one is more likely to overestimate the incidence rate of SCH, leading to excessive diagnosis and treatment.

## INTRODUCTION

With the increase in life expectancy and the aging of the population, thyroid diseases have become one of the common diseases of senior citizens. The thyroid gland, consisting of two connected lobes, is one of the largest endocrine glands in the human body, weighing 20–30 g in adults. Thyroid lesions are often found on the gland, with a prevalence of 4%–7% (*Gharib et al., 2010*). Most of them are asymptomatic, and thyroid hormone secretion is normal. Domestic epidemiological investigation shows that the overall prevalence rate of thyroid disease is 50.96%, and the prevalence rate of senior citizens is higher than that of the general population; subclinical hypothyroidism (SCH) is the most common among senior citizens, with a prevalence rate of about 20% (*Zhai et al., 2018*). SCH can further evolve into clinical hypothyroidism, and early diagnosis of subclinical thyroid

Corresponding author
Zhipeng Li, 13568985243@163.com

dysfunction is of great clinical significance. SCH has no apparent clinical symptoms and is a disease that relies on laboratory serological tests for diagnosis. As one of the vital serum indicators for diagnosing SCH, the definition of the upper limit of the reference interval for thyroid-stimulating hormone (TSH) levels is still controversial. The epidemiological characteristics of SCH vary according to gender, age, race, and region.

Studies have found that TSH levels have a skewed distribution in healthy people, whose TSH levels are infinitely close to the upper limit of the reference interval have SCH. In actual diagnosis, a smaller proportion of patients diagnosed with SCH do not develop clinical hypothyroidism over time. Using the existing diagnostic method with the upper limit of the TSH reference value as the cut-off value will inevitably lead to "false negatives" and "false positives", and there is a particular risk of misjudgment. *Waise & Price (2009)* believe that the normal reference range of TSH levels should be analyzed individually, based on clinical epidemiological characteristics. H. *Ross et al. (2009)* used the free thyroxine (FT4) and TSH levels of healthy adults as two-dimensional variables at the same time. After logarithmic standardization, they conducted a two-dimensional probability distribution study and established a 95% reference interval for the paired values of FT4 and TSH—a "two-dimensional composite reference interval". After testing, the reference interval of two-dimensional variables can reduce the number of patients misdiagnosed as "SCH" by 14% compared with the traditional interval. This method suggests a more accurate model for diagnosing SCH (*Ross et al., 2009*).

Baseline data on the epidemiological characteristics of subclinical thyroid dysfunction among senior citizens in Chengdu, Sichuan Province, China are incomplete. This study intended to establish the baseline data of SCH in the Han nationality elderly in Chengdu, exploring the effects of gender, age, blood lipid levels, *et al.* on TSH levels in the elderly with SCH. In addition, our study has established a two-dimensional compound reference interval of TSH and FT4, comparing the sensitivity and accuracy between the compound reference one and traditional univariate reference interval in diagnosing SCH in the elderly.

## MATERIAL AND METHODS

### Exploring the effects of gender, age, thyroid antibody, blood lipids, *etc.* on TSH levels in elderly SCH patients in Chengdu, China

#### *Definition and diagnostic criteria for subclinical thyroid dysfunction*

Subclinical thyroid dysfunction includes subclinical hyperthyroidism and SCH. The specific criteria are as follows:

Subclinical hyperthyroidism group: TSH is lower than the reference range; FT4 is within the reference range; Free triiodothyronine (FT3) is within the reference range.

SCH group: TSH is higher than the reference range; FT4 is within the reference range.

#### *Population inclusion and exclusion criteria*

Inclusion criteria: the patient who is self-reported healthy Han elderly aged 60 and above in Sichuan.

Exclusion criteria: the patient who is diagnosed with thyroid disease. The patient who has taken thyroid hormone or related drugs. The patient who has a family history of thyroid

disease. The patient who has a history of thyroid trauma or surgery. The patient who has a mental illness and related family history. The patient who has a malignant tumor. The patient has been on a low-iodine diet or living in low-iodine areas for a long time.

Grouping criteria: Our patients were divided into three groups according to age: junior-old (60–69 years old), middle-old (70–79 years old), and oldest-old(80 years old and above).

### Data collection

This study included patients who underwent physical examination at the Health Management Center of West China Hospital of Sichuan University from January 2018 to December 2022. General clinical data were recorded, which include age, sex, height, weight, waist circumference, hip circumference, waist-to-hip ratio, blood pressure ,etc. The laboratory test results were collected, including blood routine (white blood cell count, red blood cell count, hemoglobin, mean erythrocyte content, mean red blood cell volume, platelet count), blood biochemistry (alanine aminotransferase, aspartate aminotransferase, albumin, cholesterol, high-density lipoprotein, low-density lipoprotein, triglyceride, glucose, creatinine, urea, uric acid, estimated glomerular filtration rate, homocysteine, cystatin C, potassium, sodium, chlorine), glycosylated hemoglobin, blood parathyroid hormone, blood TSH, triiodothyronine (T3), FT3, total free thyroxine (T4), FT4, anti-thyroglobulin antibody (TgAb), anti-thyroid peroxidase antibody (TPOAb), qualitative determination of urinary protein, urinary white blood cells, and red blood cells. Electrochemiluminescence immunoassay was used for TSH, FT4 , and TPOAb. The reference range of TSH was 0.27−4.20 mIU/L, the reference range of FT4 was 12.0–22.0 pmol/L, and TPOAb (the reference range is less than 34 IU/ml) TPOAb<34 IU/L was negative. TPOAb ≥34 IU/L was positive.

This research used a retrospective study design. It was approved by the biomedical ethics review committee of the West China Hospital, Sichuan University(Trial No. 293, 2023). Informed consent was waived by the institutional review board.

### Statistical method

Descriptive statistical analysis was used to test and judge the distribution type of the above index values of the included population. The basic ideas and methods of principal component analysis were used to screen out the components that may be associated with TSH levels and cause statistically significant effects. Multiple linear regression analysis was performed with the items selected by the principal component analysis as the independent variable and TSH level as the dependent variable to explore the direction and degree of influence of related factors on TSH. Statistic Package for Social Science (SPSS) was used for data statistics and analysis, and the statistical graphics software GraphPad Prism was used for image analysis.

## Establishing two-dimensional composite reference intervals for TSH and FT4 in the elderly
### Inclusion and exclusion criteria

Inclusion criteria: Elderly people with normal TPOAb levels were selected from the first part of the included population (1,194 people).

Exclusion criteria: Same as 'Population inclusion and exclusion criteria'

### Establishment of TSH and FT4 composite reference intervals

Ross et al. (2009) found that the two-dimensional scatter plot analysis of the original data of TSH and FT4 levels (Fig. 1) showed that the upper and lower limits of the standard reference interval of TSH and FT4 divided the whole region into uneven parts. Except for the cross area of the two reference values with relatively concentrated scatter points, the probability density distribution of scatter points in the other parts was not uniform. The original data of TSH and FT4 levels were log-transformed and standardized to a standard normal distribution (Fig. 2). At this time, the spatial position of the transformed data was defined again by Mahalanobis distance. Then the scatter analysis was performed (the figure is a regular circle centered at the point (0,0)). The new composite reference interval can be obtained after the distribution probability is determined and the reference limit is defined (Ross et al., 2009). A two-dimensional composite reference interval for TSH and FT4 is established using this method.

### Statistical method

The Kolmogorov–Smirnov test was used to test the distribution of two-dimensional composite reference intervals to confirm whether they were in line with multivariate normal distribution.

## The sensitivity and accuracy of the established two-dimensional composite reference intervals were compared with those of the traditional univariate reference intervals in diagnosing SCH

In this study, two reference intervals were used to diagnose SCH. One was the traditional standard, the other was the two-dimensional composite reference interval. The number of SCH patients was obtained by the two reference intervals. In addition, the differences in diagnosing were also compared.

# RESULTS

## TSH level and its influencing factors in the elderly
### Basic information about the subjects

A total of 1,322 self-reported healthy elderly Han Chinese aged ≥60 years in Sichuan province were included. There were 759 males (57.4%) and 563 females (1.3:1). According to age, they were divided into three groups, including 915 (69.2%) in the junior-old group, 309 (23.4%) in the middle-old group, and 98 (7.4%) in the oldest-old group. There were 1,194 patients in TPOAb-negative group (90.3%), and 128 patients in the TPOAb-positive group (9.7%). There were 99 (7.5%) patients in the TgAb-positive group, and 1,220 cases in the TgAb-negative one (92.5%). There were seven patients with subclinical hyperthyroidism

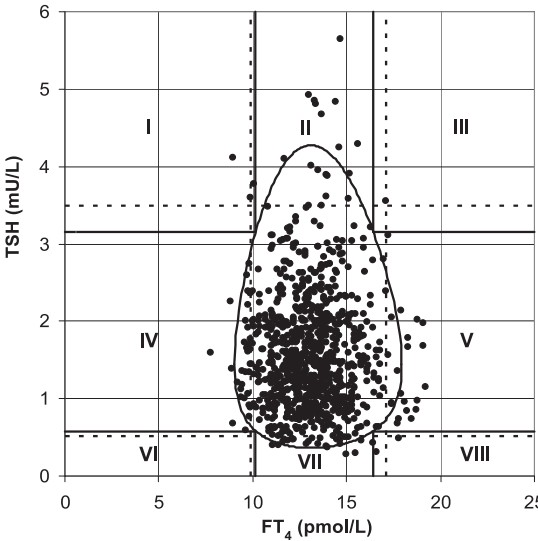

**Figure 1  TSH and FT4 values in the the traditional reference interval.**

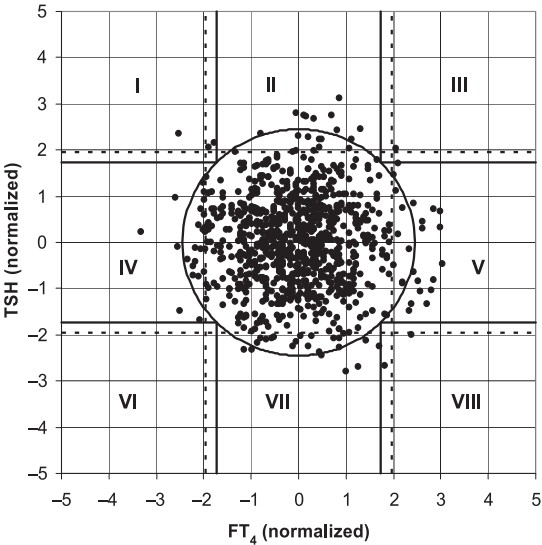

**Figure 2  TSH and FT4 values in the two-dimensional composite reference interval.**

(0.5%), and 183 patients with SCH (107 males, 76 females) (13.8%). Seventeen patients with TPOAb-positive in the SCH group, (9.6%). There were 17 patients with SCH in the TPOAb-positive group likewise (13.3%).

The mean age and waist-to-hip ratio were 67.6 and 0.89. The mean body mass index and blood pressure were 24.1 kg/ (m$^2$) and 132/74mmHg. The mean low-density lipoprotein was 3.04 mmol/L. The median of cholesterol and high-density lipoprotein were 4.76 (2.27−9.01) mmol/L and 1.34 (0.67−3.27) mmol/L. The median of triglyceride and glycosylated hemoglobin were 1.1 (0.31–15.18) mmol/L and 5.5 (3.6–13.3) %. The median

values of creatinine, urea, uric acid, and TSH were 71.0 (25.0–191.0) umol/L, 4.60 (2.1–18.2) mmol/L, 324 (110-672) umol/L, and 2.35 (0.01–12.74) mU/L, respectively. The median values of T3, FT3, and T4 were 1.76 (0.65−5.06) nmol/L, 5.05 (1.82–18.07) pmol/L, and 97.83 (23.31–227.0) nmol/L, respectively. The median values of FT4, TgAb, and TPOAb were 16.8 (3.32–43.75) pmol/L, 11.9 (10.0–4000.0) IU/ml, and 11.36 (5.0–600.0) IU/ml, respectively.

### *Comparison of general data among junior-old, middle-old, and oldest-old group*

It was found that there were statistically significant differences in height, weight, waist circumference, waist-hip ratio, systolic blood pressure, and diastolic blood pressure among the three groups (Table 1). Height, weight, and diastolic blood pressure showed a downward trend with age progression. However, waist circumference, waist-hip ratio, and systolic blood pressure showed an upward trend. The comparison of gender differences between groups was performed through $\chi^2$ test. The analysis found that there were statistically significant differences in gender among the three groups ($p < 0.05$), and there were more males than females in the three groups (Table 1).

## The differences of TSH, FT3, and FT4 among junior-old, middle-old, and oldest-old group

The analysis found that there were no statistically significant differences in TSH, FT3, and FT4 among the three groups.

## Comparison of differences between the SCH group and regular control group

We compared the intergroup differences between the normal thyroid function group(843 cases) and the SCH group (183 cases). We found that there were significant differences in creatinine, uric acid, estimated glomerular filtration rate (eGFR), TSH, FT3, T4, FT4, and TPOAb levels between the two groups ($P < 0.05$), using the Wilcoxon rank sum test. What's more, the levels of creatinine, uric acid, eGFR, FT3, T4, and FT4 in the SCH group were lower than those in the regular group. The levels of TSH and TPOAb were higher than those in the regular one (Table 2).

### *Correlation analysis between TSH and other indicators*

After analysis of the whole population, it was found that FT3, FT4, creatinine, and uric acid were related to TSH. TSH was negatively correlated with FT3, FT4, creatinine, and uric acid, all of which were statistically significant ($P < 0.05$)(Table 3).

The correlation between TSH and other indicators in the junior-old,middle-old, and oldest-old groups were further analyzed accordingly. TSH was negatively correlated with FT3, FT4, creatinine, and uric acid in the junior-old group (Table 4); TSH was positively correlated with urea. TSH was negatively correlated with FT4 and aspartate aminotransferase in the middle-old group (Table 5);  TSH was positively correlated with height. TSH was negatively correlated with the waist-hip ratio in the oldest-old group (Table 6).

**Table 1  Group comparison of general data.**

| | Elderly | $\bar{X} \pm S$ | 95% CI | P |
|---|---|---|---|---|
| Height (cm) | Young | 160.751 ± 7.9839 | 160.228, 161.275 | |
| | Middle-aged | 159.449 ± 8.2215 | 158.519, 160.378 | 0.005 |
| | Elderly | 158.425 ± 8.8365 | 156.542, 160.309 | |
| Weight (kg) | Young | 62.158 ± 10.0479 | 61.500, 62.816 | |
| | Middle-aged | 61.514 ± 10.9392 | 60.277, 62.750 | 0.006 |
| | Elderly | 58.508 ± 9.8620 | 56.406, 60.610 | |
| Waistline (cm) | Young | 83.274 ± 9.0537 | 82.681, 83.868 | |
| | Middle-aged | 84.782 ± 9.2528 | 83.736, 85.828 | 0.027 |
| | Elderly | 84.655 ± 8.9258 | 82.753, 86.558 | |
| Waist hip rate | Young | 0.8851 ± 0.06384 | 0.8809, 0.8893 | |
| | Middle-aged | 0.9016 ± 0.06723 | 0.8940, 0.9093 | 0.000 |
| | Elderly | 0.9082 ± 0.05970 | 0.8954, 0.9209 | |
| Systolic pressure (mmHg) | Young | 129.70 ± 17.100 | 128.58, 130.82 | |
| | Middle-aged | 135.73 ± 18.112 | 133.69, 137.77 | 0.000 |
| | Elderly | 138.87 ± 17.516 | 135.24, 142.50 | |
| Diastolic blood pressure (mmHg) | Young | 75.16 ± 10.660 | 74.46, 75.86 | |
| | Middle-aged | 72.77 ± 10.785 | 71.55, 73.98 | 0.000 |
| | Elderly | 68.75 ± 10.776 | 66.52, 70.98 | |

**Table 2  Comparison of the differences between the subclinical hypothyroidism group and the normal control group.**

| Index (unit) | SCH group M (P25, P75) | normal group M (P25, P75) | Z-value | P (two-tailed) |
|---|---|---|---|---|
| Creatinine (umol/L) | 67 (57.00, 81.00) | 71.00 (60.00, 85.00) | −2.660 | 0.008 |
| Uric Acid (umol/L) | 306.00 (261.00, 374.00) | 324.00 (268.00, 395.00) | −2.486 | 0.013 |
| Estimated glomerular filtration rate (ml/min/1.73m$^2$ ) | 94.83 (80.83, 107.4) | 99.78 (87.53, 110.47) | −2.507 | 0.012 |
| Thyroid-stimulating hormone (mIU/L) | 5.06 (4.64, 6.18) | 2.35 (1.65, 3.47) | −21.228 | 0.000 |
| Free triiodothyronine (pmol/L) | 4.93 (4.56, 5.38) | 5.05 (4.64, 5.50) | −3.042 | 0.002 |
| Thyroxine (nmol/L) | 16.15 (14.96, 17.53) | 97.83 (86.76, 108.00) | −2.027 | 0.043 |
| Free thyroxine (pmol/L) | 16.15 (14.96, 17.53) | 16.80 (15.36, 18.37) | −4.012 | 0.000 |
| Anti-thyroid peroxidase antibody (IU/ml) | 12.07 (9.00, 17.17) | 11.36 (9.00, 16.803) | −3.553 | 0.000 |

Spearman's correlation coefficient was used to compare the correlation between TSH and other indicators in 183 patients in the SCH group (Table 7). TSH was positively correlated with height, aspartate aminotransferase, and albumin. It was negatively correlated with uric acid with statistical significance ($P < 0.05$).

### Multiple linear regression analysis

Multiple linear regression analysis was performed with TSH as the dependent variable and variables related to TSH as independent variables (Table 8).

The results showed that in the multiple linear regression analysis of TSH, the variables entering the regression model were FT4 and uric acid. The multiple linear regression equation was TSH = −0.099 × FT4-0.001 × uric acid+4.875. When FT4 increases by

**Table 3  Correlation analysis of whole population TSH and other indicators.**

| Whole population | | | TSH |
|---|---|---|---|
| Spearman | FT3 | Correlation index | $-.105^{**}$ |
| | | Significance (two-tailed) | 0.000 |
| | | Number of cases | 1321 |
| | FT4 | Correlation index | $-.148^{**}$ |
| | | Significance (two-tailed) | 0.000 |
| | | Number of cases | 1321 |
| | Creatinine | Correlation index | $-.105^{**}$ |
| | | Significance (two-tailed) | 0.000 |
| | | Number of cases | 1320 |
| | Uric Acid | Correlation index | $-.099^{**}$ |
| | | Significance (two-tailed) | 0.000 |
| | | Number of cases | 1320 |

Notes.

[**]At level 0.01 (two-tailed), the correlation was significant.

**Table 4  Correlation analysis of TSH in junior-old group and other indicators.**

| Junior-old group | | | TSH |
|---|---|---|---|
| Spearman | FT3 | Correlation index | $-.118^{**}$ |
| | | Significance (two-tailed) | 0.000 |
| | | Number of cases | 915 |
| | FT4 | Correlation index | $-.154^{**}$ |
| | | Significance (two-tailed) | 0.000 |
| | | Number of cases | 915 |
| | Creatinine | Correlation index | $-.148^{**}$ |
| | | Significance (two-tailed) | 0.000 |
| | | Number of cases | 914 |
| | Uric acid | Correlation index | $-.126^{**}$ |
| | | Significance (two-tailed) | 0.000 |
| | | Number of cases | 914 |

Notes.

[**]At level 0.01 (two-tailed), the correlation was significant.

1pmol/L, TSH decreases by 0.099mIU/L with the rest factors remained stable. Similarly, when uric acid increasing by 1umol/L, TSH decreases by 0.001mIU/L.

## Establishment of two-dimensional composite reference intervals for TSH and FT4

Junior-old group $D^2 = (\frac{\sqrt{TSH+0.386}-1.595}{0.407})^2 + (\frac{\sqrt{FT4+0.117}-4.098}{0.274})^2$

Middle-old group $D^2 = (\frac{\sqrt{TSH+0.403}-1.530}{0.368})^2 + (\frac{\sqrt{FT4+0.006}-4.112}{0.253})^2$

Oldest-old group $D^2 = (\frac{\sqrt{TSH+0.057}-1.589}{0.361})^2 + (\frac{\sqrt{FT4+0.643}-4.155}{0.273})^2$

**Table 5  Correlation analysis of TSH in middle-old group and other indicators.**

| Middle-old group | | | TSH |
|---|---|---|---|
| Spearman | FT4 | Correlation index | −.118* |
| | | Significance (two-tailed) | 0.038 |
| | | Number of cases | 308 |
| | Aspartate aminotransferase | Correlation index | −.157** |
| | | Significance (two-tailed) | 0.006 |
| | | Number of cases | 308 |
| | Urea | Correlation index | .128* |
| | | Significance (two-tailed) | 0.024 |
| | | Number of cases | 308 |

Notes.
*At level 0.01 (two-tailed), the correlation was significant.

**Table 6  Correlation analysis of TSH in oldest-old group and other indicators.**

| Oldest-old group | | | TSH |
|---|---|---|---|
| Spearman | Height | Correlation index | .240* |
| | | Significance (two-tailed) | 0.025 |
| | | Number of cases | 87 |
| | Waist hip rate | Correlation index | −.215* |
| | | Significance (two-tailed) | 0.046 |
| | | Number of cases | 87 |

Notes.
*At level 0.01 (two-tailed), the correlation was significant.

**Table 7  Correlation analysis of TSH in SCH group and other indicators.**

| SCH group | | | TSH |
|---|---|---|---|
| Spearman | Height | Correlation index | .160* |
| | | Significance (two-tailed) | 0.035 |
| | | Number of cases | 174 |
| | Aspartate aminotransferase | Correlation index | .184* |
| | | Significance (two-tailed) | 0.014 |
| | | Number of cases | 177 |
| | Albumin | Correlation index | .167* |
| | | Significance (two-tailed) | 0.027 |
| | | Number of cases | 176 |
| | Uric Acid | Correlation index | −.155* |
| | | Significance (two-tailed) | 0.039 |
| | | Number of cases | 177 |

Notes.
**At level 0.01 (two-tailed), the correlation was significant.

A unified two-dimensional composite reference interval was constructed by using the whole elderly population as the reference population:

$$D^2 = (\frac{\sqrt{TSH + 0.377} - 1.581}{0.399})^2 + (\frac{\sqrt{FT4 + 0.144} - 4.104}{0.268})^2.$$

**Table 8  Multiple linear regression analysis of TSH.**

| Index (unit) | B | SE | 95%CI | P |
|---|---|---|---|---|
| Free thyroxine (pmol/L) | −0.099 | 0.017 | −0.132 , −0.066 | 0.000 |
| Uric Acid (umol/L) | −0.001 | 0.000 | −0.002 , 0.000 | 0.004 |
| (Constant) | 4.875 | 0.314 | 4.258, 5.491 | 0.000 |

The constants 0.377 and 0.144 are the standardized skewness coefficients of TSH and FT4, respectively; 1.581 and 4.104 are the standardized mean values of TSH and FT4, respectively; 0.399 and 0.268 are the standard deviations.

Both FT4 and TSH have two corresponding values of TSH and FT4 separately, which are the ranges of TSH and FT4. For example, when FT4 is 12pmol/L, the corresponding range of TSH is 4.0−8.0. When the TSH value is 6.8mIU/L, the range of FT4 values is 15-20. When $D^2 > 5.99$, hypothyroidism was determined.

**Validation of two-dimensional composite reference intervals**

183 and 130 SCH patients were diagnosed by using the traditional and the two-dimensional composite reference intervals as the diagnostic standard, respectively. The use of two-dimensional composite reference intervals reduced the number of patients diagnosed with SCH by 4%. All patients were tested for TPOAb, and 128 were positive. Among the TPOAb-positive patients, 17 and 11 SCH patients were diagnosed by using the traditional and the two-dimensional composite reference intervals as the diagnostic standard, respectively.

When we used the two-dimensional composite reference interval formula for the junior, middle, and oldest-old diagnostic criteria accordingly, the number of diagnosing SCH was 75, 37, and nine, respectively, for a total of 121 cases. Compared with the two-dimensional composite reference interval of the whole population, the total number of patients differed by 9. After the chi-square test analysis, there is no statistically significant difference between the formula of the three groups and the whole population. Therefore, the two-dimensional composite reference interval of the whole population can be used as the unified diagnostic standard.

## DISCUSSION

Thyroid diseases, including thyroid tumors, have become one of the most common diseases in the elderly. The Bethesda Thyroid Cytopathology (BSRTC) reporting system is the most widely used of the fine needle aspiration cytology (FNAC) results reporting systems. The proportion of accidental malignancies reported in Bethesda II was 1.53% (*Mulita et al., 2022*). The incidence of occasional malignancy in multiple nodular goiter and single nodular goiter was 18.42% and 20.13%, respectively, and both were diagnosed as Bethesda III (*Mulita et al., 2021*). Lesions that produce Bethesda II and III results include some disorders with abnormal thyroid function (*Mulita et al., 2022*). SCH is also the most common thyroid disease in the elderly, with a prevalence of up to 20%. 90% of the patients have mild SCH with atypical clinical manifestations, and the diagnosis mainly depends on laboratory indicators (*Zhai et al., 2018*). *Ross et al. (2009)* found that the reference interval

of two-dimensional variables can reduce the misdiagnosis of SCH by 14% compared with the traditional interval. It is of practical value to use the two-dimensional composite reference interval of TSH and FT4 in the diagnosis of SCH in the elderly (*Ross et al., 2009*). The prevalence of subclinical thyroid dysfunction is about 14% in the elderly over 60 years old in our study, with SCH accounting for about 97%.

## TSH level and its influencing factors in elderly with subclinical thyroid dysfunction in the local area

It was found that the height, weight, and diastolic blood pressure of the elderly showed a downward trend with the increase of age. However, waist circumference, waist-hip ratio, and systolic blood pressure showed an upward trend. It is suggested that the elderly are prone to abdominal obesity and suffer from hypertension with high systolic blood pressure. *Xue et al. (2022)* included 13,251 elderly population with abdominal obesity of 951(34.34%). The present study did not find TSH levels increased with age, which is inconsistent with other studies. In addition, the sixth round of the census showed that the prevalence of hypertension in people aged 60–69 years increased slowly over time (*Zhang et al., 2023*). However, Some scholars pointed out TSH is negatively correlated with age. *Xue & Feng (2012)* included 2,443 patients, dividing the population into junior-old (<60 years old), middle-old (60–79 years old), and oldest-old (≥80 years old) groups. It was found that TSH showed a downward trend with the increase in age, and the decline rate of TSH in the oldest-old group was higher than that in the middle-old group. While some scholars believe that there is little correlation between TSH and age. *Li et al. (2011)* included 602 healthy physical examination subjects and divided them into non-old (<60 years old), junior-to-middle old (60–79 years old) and oldest-old (≥80 years old) groups. The correlation between TSH and age is low, and TSH increased with age only in the oldest-old group. Multi-center studies at home and abroad have found many factors affecting TSH, and the correlation between age and TSH cannot be considered only. The Third National Survey on Nutrition and Health (NHANES III) in the United States pointed out that the reference range of TSH should be based on age, race, and gender. After 39 years of age, the 97.5th percentile of serum TSH increased by 0.3mIU/L for every ten years of age increase (*Boucai, Hollowell & Surks, 2011*; *Zhai & Shan, 2014*). A multi-center study has found that age, gender, and region were all the influencing factors of TSH reference interval in the elderly. In addition, TSH did not increase with age in elderly women. There is a positive correlation between TSH level and age in elderly male patients, and there is a significant difference after the age of 75 (*Wang et al., 2023*). In our study, the serum TSH level did not increase with age, which may be related to the inclusion of only the elderly but not the non-elderly, as well as different regions and laboratory detection methods.

Our study found that the levels of creatinine, uric acid, eGFR, FT3, T4, and FT4 in the SCH group were lower than those in the normal group. TPOAb-positive was higher than that of normal elderly. While the rest of the clinical indexes with none of significant importance. The phenomenon reminds us that elderly patients with SCH have no obvious clinical manifestations. For instance, the renal metabolic indexes perform better in the SCH group than in the normal elderly. Existing studies are controversial on whether SCH

affects the renal system. *Gao et al. (2022)* retrospectively analyzed 103,513 subjects (18-101 years) and found that serum creatinine gradually increased with aging. In addition to renal function, the influencing factors of serum creatinine include muscle weight, gender, age, and exogenous food intake *Meuwese et al. (2019)* pointed out that there was no significant difference in renal function decline between patients with normal thyroid function and patients with SCH. However, unlike *Meuwese et al. (2019)*, *Kim et al. (2023)* showed that SCH is an independent risk factor for chronic kidney disease. The relationship between TSH and serum uric acid is controversial in existing studies. *Feng et al. (2021)* included 254 type 2 diabetic patients with early renal damage. It was found that TSH is negatively correlated with blood uric acid. *Feng et al. (2021)* further found that a decrease in TSH is an independent risk factor for subjects with hyperuricemia, speculating that the negative correlation between TSH and serum uric acid was the result of insulin resistance. *Yang & Cao (2022)* included 19,013 subjects and found that the risk of hyperuricemia was related to elevated TSH; Male patients with elevated TSH values were more likely to suffer from hyperuricemia.

Among the included elderly population in the local area, the median TSH level of SCH elderly was 5.06 (4.22–12.74) mIU/L. The correlation analysis showed that the level of TSH in the elderly increased with the decrease of FT3, FT4, creatinine, and uric acid. Multiple regression analysis showed that serum FT4 and uric acid were the most critical factors affecting TSH in elderly patients with SCH. On the one hand, the secretion of TSH is affected by the promotion of TRH secreted by the hypothalamus; on the other hand, it is also affected by the feedback inhibition of T3 and T4. The two regulate each other, and they form the hypothalamic-adenopituitary-thyroid axis. Under normal circumstances, there is a negative correlation between FT4 and TSH to maintain a relatively stable level of thyroid hormone. Like the present study, *Hoermann et al. (2010)* included subjects with different thyroid functions and confirmed the negative correlation between TSH and FT4. *De Grande, Van Uytfanghe & Thienpont (2015)* also observed a negative correlation between TSH and FT4. They included 8152 patients (18–100 years old) and divided the subjects into three groups of low, medium, and high concentration according to TSH levels, and found that there was a negative correlation between TSH and FT4 in all three groups. *Benhadi et al. (2010)* enrolled 21 healthy volunteers (average age 60 years old), and randomly assigned them to receive an oral placebo, 125 mg T4, and 250 mg T4; Fasting blood samples were collected from volunteers before and after administration. The study found that after administering T4, serum FT4 levels increased while TSH levels decreased, indicating a negative correlation between FT4 and TSH (*Benhadi et al., 2010*). In clinical practice, patients with elevated TSH should pay attention to screening for any decrease in blood FT4 values and uric acid levels to avoid missed diagnosis.

## Establishment and verification of two-dimensional composite reference intervals for TSH and FT4 in the elderly

In this study, 1,194 subjects with normal TPOAb levels were screened out from 1,322 elderly people and included in the study to establish a two-dimensional composite reference

intervals Finally, the two-dimensional composite reference interval of the elderly population was constructed.

At present, there are few reports on the establishment of two-dimensional composite reference intervals for TSH and FT4 in the elderly. In this study, the traditional reference interval and two-dimensional composite reference interval were used to diagnose SCH, and the difference in the diagnostic rate of the two methods was compared. The two-dimensional composite reference interval could reduce 4.0% of the patients diagnosed as "SCH" compared with the traditional reference interval. This difference was as high as 4.7% in TPOAb-positive subjects. This is consistent with the demonstration results of the "two-dimensional composite reference interval" established by *Ross et al. (2009)*, as well as our previous research (*Liang et al., 2014*). The traditional diagnosis of SCH is combined and classified by the cut-off values corresponding to the 95% reference intervals of FT4 and TSH in the healthy reference population. Even if a report with TSH, FT3, and FT4 is obtained, it is interpreted separately, which may increase the probability of misdiagnosis. Several studies have shown that the relationship between TSH and FT4 is not a simple linear correlation. *Hoermann et al. (2010)* included subjects with different thyroid function statuses (hypothyroidism, normal, and hyperthyroidism), introducing a nonlinear mathematical model. It was found that the relationship between TSH and FT4 is not a simple linear logarithm (*Hoermann et al., 2010*). *Ross et al. (2009)* took FT4 and TSH levels of healthy adults as two-dimensional variables at the same time, and conducted two-dimensional probability distribution research after logarithmic normalization, and established the 95% reference interval of FT4 and TSH paired values. Similarly, *Hoermann et al. (2016)* established a two-dimensional composite reference system based on 271 healthy subjects, validating it in 820 untreated subjects with thyroid disease. The results showed that 26% of the patients diagnosed with "thyroid dysfunction" according to the traditional criteria were "euthyroid" according to the two-dimensional composite reference system (*Hoermann et al., 2016*). A two-dimensional composite reference interval was used to represent the association between TSH and FT4 in our study. The distribution of FT4 in healthy people was approximately normal, while the distribution of TSH was skewed. After logarithmic transformation, both FT4 and TSH showed normal distribution. The scatter distribution of TSH and FT4 in the traditional reference interval is similar to a duck egg, rather than a regular circle. Thus, the traditional reference interval increases the operability and difficulty of accurately measuring 95% of the data volume, which may increase the accuracy and difficulty of diagnosing diseases in this reference interval. The transformation was fine-tuned by correcting residual skewness, and FT4 was log-transformed and normalized to derive a regular circular two-dimensional composite reference interval, which was convenient for observation and diagnosis of 95% normal population distribution. Compared with the established two-dimensional composite reference interval, the traditional reference interval may be more likely to overestimate the incidence of SCH in the elderly, resulting in overdiagnosis and treatment. We should also notice another possibility that the two-dimensional composite reference interval may lead to the "underdiagnosis" of SCH. It indicates that clinicians should be careful to diagnose SCH with dynamic follow-up of thyroid function. The sample size of this study is relatively

small, which affects the accuracy and stability of the research conclusions. In the future, the sample size will be increased to verify the reliability and validity of the two-dimensional composite reference interval.

## CONCLUSION

1. The prevalence of subclinical thyroid dysfunction in the elderly population in this region is about 14%, among which SCH accounts for about 97%. In the elderly population, the positive rate of TPOAb and TgAb is about 10% and 8%, respectively.

2. Multiple regression analysis showed that TSH in the elderly tended to increase with the decrease of serum FT4 and uric acid levels.

3. Compared with the two-dimensional composite reference intervals established after the standard transformation of TSH and FT4, the traditional independent reference intervals may overestimate the incidence of SCH in the elderly, leading to overdiagnosis and treatment.

### Funding

This research was funded by a grant from the Key Project of Science and Technology Department of Sichuan Province (No. 2023YFG0278) and the Health science research project of Sichuan province (No. 2023-108). The funders had no role in study design, data collection and analysis, decision to publish, or preparation of the manuscript.

### Grant Disclosures

The following grant information was disclosed by the authors:
The Key Project of Science and Technology Department of Sichuan Province: 2023YFG0278.
The Health science research project of Sichuan province: 2023-108.

### Competing Interests

The authors declare there are no competing interests.

### Author Contributions

- Peijuan Li conceived and designed the experiments, performed the experiments, analyzed the data, prepared figures and/or tables, authored or reviewed drafts of the article, and approved the final draft.
- Wenming Yang performed the experiments, analyzed the data, prepared figures and/or tables, authored or reviewed drafts of the article, and approved the final draft.
- Guohua Tang performed the experiments, analyzed the data, prepared figures and/or tables, authored or reviewed drafts of the article, and approved the final draft.
- Zhipeng Li conceived and designed the experiments, performed the experiments, analyzed the data, prepared figures and/or tables, authored or reviewed drafts of the article, and approved the final draft.

## Human Ethics

The following information was supplied relating to ethical approvals (*i.e.*, approving body and any reference numbers):

West China Hospital, Sichuan University granted Ethical approval to carry out the study within its facilities (Ethical Application Ref: Trial No. 293, 2023).

## Data Availability

The raw measurements are available in the Supplemental Files.

## Supplemental Information

Supplemental information for this article can be found online at http://dx.doi.org/10.7717/peerj.18417#supplemental-information.

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
