# Peer review of "Application of composite reference intervals in the diagnosis of subclinical hypothyroidism in the elderly: a retrospective study"

_PeerJ, doi:10.7717/peerj.18417_

## Round 0.1 · original submission · Minor Revisions

· Academic Editor

Minor Revisions

Dear authors,

The study entitled “Application of composite reference intervals in the diagnosis of subclinical hypothyroidism in the elderly” demonstrated interesting findings. However, minor revisions must be clarified in the manuscript.

Reviewer 1 ·

Basic reporting

No comment

Experimental design

No comment

Validity of the findings

No comment

Additional comments

in line 305-307: It was found that TSH showed a downward trend with the increase in age, and the decline rate of TSH in the elderly group was higher than that in the elderly group. This needs rectification

Reviewer 2 ·

Basic reporting

This is good article

Experimental design

I am satisfied with this design

Validity of the findings

I think this is fair enough accepted finding

Additional comments

No more comments

·

Basic reporting

1) "The thyroid gland, consisting of two connected lobes, is one of the largest endocrine glands in the human body, weighing 20 - 30 g in adults. Thyroid lesions are often found on the gland, with a prevalence of 4%–7%. Most of them are asymptomatic, and thyroid hormone secretion is normal."
I would suggest adding this information in the introduction section

Experimental design

2) I would like a brief discussion on the Bethesda classification system for reporting thyroid cytopathology ( especially for type II and III) and consider citing the recently published articles on Bethesda II and III:
https://pubmed.ncbi.nlm.nih.gov/33749812/
https://pubmed.ncbi.nlm.nih.gov/34734516/

What is the percentage of incidental malignancy according to these studies for Bethesda II and III?

Validity of the findings

Methods
- The methods are sufficiently explained by the authors.

Results
- The results are presented in a very extensive way.

Additional comments

Discussion
- The discussion is of great quality and includes updated data.
Conclusion
From the presented data, the conclusion is complete and represents the work that the authors did.

Reviewer 4 ·

Basic reporting

1. Overall, the manuscript is redundant and hard to read, especially the Results and Discussion sections. In the Discussion section, the authors should summarize their findings, rather than rewriting them.

2. Is data of Figure 1 and 2 in the Materials and methods section cited from ref 3? If the data is original, they should be written in the Results section.

Experimental design

1. If the authors think SCH may lead to renal failure and increase in UA (rather than renal failure and high UA causing SCH), they should use correlation analysis instead of regression analysis.

Validity of the findings

1. It seems to the reviewer that some necessary data is missing. Where is the result of principal component analysis? If extra data is listed in Supplemental Data, it should be stated accordingly in the manuscript.

2. If the authors want to claim that the two-dimensional composite reference interval they established are more efficient in diagnosing SCH than using the traditional reference interval, they should compare the clinical characteristics of the 4% with “excessive diagnosis” to the rest of the 96% SCH. It is possible that the two-dimensional composite reference interval leading to “underdiagnosis” of SCH.

3. The words in Table 2 are not aligned. Factors affecting UA or S-Cr, such as BMI and glycosylated hemoglobin, should also be compared between SCH group and normal group.

---

## Round 0.2 · accepted · Accept

· Academic Editor

Accept

Dear Author,

Congratulations, after the good work of revisions in response to the reviewers' comments, I would like to inform you that your manuscript has been accepted for publication in PeerJ.

·

Basic reporting

All requested changes were performed. The manuscript can be accepted for publication without further corrections.

Experimental design

All requested changes were performed. The manuscript can be accepted for publication without further corrections.

Validity of the findings

All requested changes were performed. The manuscript can be accepted for publication without further corrections.

Additional comments

All requested changes were performed. The manuscript can be accepted for publication without further corrections.